# The New ELN Recommendations for Treating CML

**DOI:** 10.3390/jcm9113671

**Published:** 2020-11-16

**Authors:** Rüdiger Hehlmann

**Affiliations:** ELN-Foundation, Weinheim and Medical Faculty Mannheim of Heidelberg University, 69126 Mannheim, Germany; hehlmann.eln@gmail.com

**Keywords:** chronic myeloid leukemia, ELN recommendations, diagnosis, monitoring, treatment, treatment-free remission, blast crisis

## Abstract

After normal survival has been achieved in most patients with chronic myeloid leukemia (CML), a new goal for treating CML is survival at good quality of life, with treatment discontinuation in sustained deep molecular response (DMR; MR^4^ or deeper) and treatment-free remission (TFR). Four tyrosine kinase inhibitors (TKIs) have been approved for first-line therapy: imatinib, dasatinib, nilotinib, bosutinib. Unexpectedly, the outcome of long-term randomized trials has shown that faster response as achieved by higher doses of imatinib, imatinib in combination, or second-generation (2G)-TKIs, does not translate into a survival advantage. Serious and frequent, and in part cumulative long-term toxicities, have led to a reevaluation of the role of 2G-TKIs in first-line therapy. Generic imatinib is the current most cost-effective first-line therapy in the chronic phase. A change of treatment is recommended when intolerance cannot be ameliorated or molecular milestones are not reached. Patient comorbidities and contraindications of all TKIs must be considered. Risk profile at diagnosis should be assessed with the EUTOS score for long-term survival (ELTS). Monitoring of response is by polymerase chain reaction (PCR). Cytogenetics is still required in the case of atypical translocations, atypical transcripts, and additional chromosomal aberrations. TKIs are contraindicated during pregnancy. Since the majority of patients are at risk of lifelong exposure to TKIs, amelioration of chronic low-grade side effects is important.

## 1. Introduction

Over the last two decades, clinical trials and population-based registries have shown that the vast majority of chronic myeloid leukemia (CML) patients treated with the tyrosine kinase inhibitor (TKI) imatinib have achieved normal life expectancy [1,2,3,4,5]. The European LeukemiaNet (ELN) has accounted for this development with management recommendations in 2006, 2009, and 2013 [6,7,8].

However, expectations of a general cure for CML remain unmet. Most patients have residual molecular disease, are not yet cured, and require life-long therapy. CML did not become the expected model disease for targeting other leukemias or cancers with TKIs, but the elucidation of pathogenesis for successfully treating cancer has been convincingly demonstrated in CML.

Some patients, who stop treatment in sustained deep molecular remission (DMR), achieve durable treatment-free remissions (TFRs) and may be cured. Treatment discontinuation and TFRs have become an important new treatment goal of CML. Duration of DMR (MR^4^ or deeper) may be the best predictor of success. A review of CML, spanning from its first recognition to its current progress, has recently been published [9]. 

To reflect this new situation, the ELN has revised and updated its recommendations for treating CML [10]. This article will review, and in some instances update, these recommendations.

## 2. Initial Diagnostic Workup

At baseline, a complete blood cell count with differential should be obtained and a physical examination should be conducted, paying special attention to spleen and liver size. A bone marrow aspirate is recommended, as is a cytogenetic analysis for the verification of the Philadelphia (Ph)-chromosome. A qualitative polymerase chain reaction (PCR) is required to identify BCR-ABL1 transcripts and transcript types. An electrocardiogram, standard clinical chemistry, and a hepatitis serology are recommended for the recognition of comorbidities [10].

Risk score at diagnosis: In the TKI era, evaluation of risk at diagnosis should be done with the EUTOS score for long-term survival (ELTS), since it predicts death by CML more accurately than the other scores [11,12]. The ELTS score uses the same variables as the Sokal score, but with different weights (Table 1). Age has become a much less important factor, since in the TKI era death by CML is less dependent on age. The numbers in red highlight the most relevant differences between Sokal and ELTS scores.

## 3. Treatment Monitoring

Monitoring of treatment is now almost exclusively done molecularly by quantitative PCR, according to the International Scale (IS) [13,14]. Molecular monitoring of treatment with imatinib was systematically undertaken by CML study IV since its beginning in 2002 [2,15,16] and resulted in the recognition of the impact of early response on outcome and of benchmark times for deep molecular remission (DMR). PCR results of the IRIS study served to define response milestones based on the IS [7,8,17]. BCR-ABL1 ≤1% was determined to be equivalent to complete cytogenetic remission [18]. Results of quantitative PCR depend on the ability of laboratories to measure absolute numbers of control-gene transcripts (ABL1, GUS), and to achieve the PCR sensitivity required for BCR-ABL1 detection. The updated milestone proposal by the ELN is shown in Table 2.

Response milestones are the same for first- and second-line therapy. Cytogenetics is no longer recommended for routine monitoring. A cytogenetic analysis with bone marrow puncture is indicated if molecular monitoring is not possible, as in the case of atypical translocations, atypical transcripts, or additional chromosomal aberrations (ACAs) [10].

For patients aiming at TFR, BCR-ABL1 < 0.01% (<MR^4^) at any time is considered to be the optimal response. 

The ELN recommendations accept a change of treatment if major molecular remission (MMR) is not reached by 36–48 months [10]. 

## 4. First-Line Treatment

First-line treatment continues to be a TKI. Meanwhile, 4 TKIs are approved for first-line use: imatinib (Glivec, Novartis), dasatinib (Sprycel, BMS), nilotinib (Tasigna, Novartis), and now also bosutinib (Bosulif, Pfizer). Radotinib (Supect, Dae Wong Pharma) is approved in South Korea only and is not discussed further in this review.

Imatinib: The first-generation TKI imatinib was compared to conventional IFN-based therapy in the IRIS study and showed superior cytogenetic and molecular response rates and a survival advantage [17]. Tolerability was better with imatinib than with IFN. The most frequent side effects were fluid retention, gastro-intestinal symptoms, and fatigue [17,19,20]. No serious toxicity has surfaced in more than 20 years of use. Patients with low cardiac ejection fractions and reduced glomerular filtration rates need to be observed more carefully [20].

Five- and ten-year data of randomized trials with imatinib combined with IFN or low-dose cytarabine, with imatinib in a higher dosage, or with the second-generation TKIs (2G-TKIs) dasatinib and nilotinib, have shown that faster responses do not translate into better survival rates than treatment with imatinib alone at the standard dose of 400 mg daily [2,21,22]. 

Generic imatinib, now available worldwide, has been determined to be a cost-effective initial treatment in chronic phase (CP) CML in several independent evaluations [23,24,25]. 

Dasatinib: This 2G-TKI inhibits the imatinib-resistant BCR-ABL1 kinase domain (KD)-mutations Y253H, E255V/K, and F359V/I/C, and induces responses faster than imatinib. Regarding the pleuro-pulmonary toxicity of dasatinib, a history of pleura and lung diseases represents a contraindication for dasatinib [20,21]. The approved dose is 100 mg/day in CPs and 140 mg/day in advanced phases. A more tolerable dose of 50 mg/day also seems to be effective in CPs [26].

Nilotinib: Nilotinib is another 2G-TKI that inhibits the imatinib-resistant BCR-ABL1 KD mutations F317L/VLI/C, T315A, and V299L, and induces faster responses than imatinib. Regarding the cardio-vascular toxicity of nilotinib, a history of coronary heart disease, cerebral-vascular events, or peripheral arterial obstructive disease represent contraindications for nilotinib. Hypertension, diabetes mellitus, hypercholesterolemia, and a history of pancreatitis are relative contraindications [20,22,27]. As a first-line treatment, the approved dose is 300 mg twice daily.

The lack of a survival advantage, and the serious and in part cumulative long-term toxicity, have led to the re-evaluation of 2G-TKIs for use in first-line therapy. Comorbidities of patients [28] and long-term toxicities of TKIs need to be carefully considered [19,20].

Bosutinib: Bosutinib is a third 2G-TKI which recognizes imatinib-resistant BCR-ABL1 KD mutations, and also induces faster responses than imatinib. No relevant comorbidities have been detected as contraindications for bosutinib [29]. However, no long-term data are available yet. Frequently, annoying but usually self-limited diarrhea occurs. If needed, treatment with loperamide is recommended. The approved dose is 400 mg/day in first-line treatments.

## 5. Second- and Higher-Line Treatment

Second-line therapy, after intolerance or resistance to the first-line TKI, is usually also a TKI. If resistance is suspected, compliance should be assessed first. If resistance is confirmed, a mutational analysis should be initiated, and treatment changed on clinical grounds without waiting for the mutation results. Resistant BCR-ABL1 KD-mutations account for about a third of cases with resistance in CPs, and are relatively rare, occurring in 4.6% in 1536 CP patients over 10 years [2]. Resistant mutations are more frequent in advanced phases. A definite choice of the second-line TKI is made according to the mutational analysis and the patient’s comorbidities. An allogeneic transplantation must be considered, and a donor search initiated.

BCR-ABL1 KD mutations are detectable with sensitivities of about 20% by Sanger sequencing, and of about 3% with next-generation sequencing (NGS) [30,31]. NGS is the recommended technology for the detection of resistant BCR-ABL1 mutations. 

Ponatinib: Ponatinib, a third-generation TKI, is still the only approved TKI with efficacy against the resistant T315I mutation. After resistance to two TKIs, treatment should be changed to another 2G-TKI or to ponatinib [32]. Because of the cardio-vascular toxicity of ponatinib, a dose reduction from 45 to 15 mg/day after a response is achieved may be appropriate. Evidence for this reduction is provided by preliminary results from the randomized dose-optimization study Optic, comparing efficacy and tolerability of ponatinib at 45, 30, and 15 mg/day [33]. In the case of a T315I mutation, or of compound mutations or progression, ponatinib at a dose of 45 mg/day should be given immediately, with reduction to 15 mg/day after response. 

All treatment decisions should be made by shared decision-making [34], after full information is given to the patient on the efficacy and side effects of the respective treatments, including transplantation, the option of treatment discontinuation, and TFR.

Allogeneic stem cell transplantation: Transplantation has retained an important place in CML treatment. Early transplantation in the CP is associated with better outcomes than late transplant [35]. Transplant-related mortality is low in CPs, but graft-versus-host disease (GvHD) remains a problem. In the TKI era, transplantation has shifted from CP to blast crisis (BC). In a report from the German CML Study Group, 6-year survival of 28 imatinib-pretreated patients in advanced phase, and 25 in blast crisis, was 49% [36,37]. Similar data were reported from China [38] and by another German group [39]. Most long-time survivors after BC had undergone transplantation (Figure 1), making transplantation a preferred treatment modality in BC. Early transplantation also seems to be more successful when high-risk additional chromosomal aberrations (ACAs) are present [40]. 

## 6. CML End-Phase and Blast Crisis

The outcome of patients in BC has not much improved. Survival is generally less than one year, and new approaches are needed. Earlier recognition of BC by genetic assessment (ACAs, mutations) might improve outcomes [40,41,42,43,44,45]. End-phase CML comprises early progression with emerging high-risk ACAs, and late progression with failing hematopoiesis and blast increase. Figure 2 illustrates the current understanding of CML progression and of ABL1 activation by radical oxygen species (ROS) [46], and/or polycomb repressive complex (PRC) [47] driven epigenetic reprogramming [48].

Chromosomal aberrations, in addition to the Ph-chromosome [40,41,42,43,44,45,49,50] and BCR-ABL1 KD mutations [51], are observed in up to 90% and 80% of BC patients, respectively. Additionally, somatic mutations are reported in BC and are associated with poor-risk disease [44,45]. Mutated genes include RUNX1, ASXL1, and IKZF1. High-risk ACAs comprise the most frequently observed ACAs in BC, that is, +8, +Ph, i(17q), +19, +21, +17 (major route ACA), some minor route ACAs (-7/7q-, 3q26.2, and 11q23 rearrangements), and complex aberrant karyotypes [49,50]. High-risk ACAs at low blast counts herald death by CML [40]. The emergence of high-risk ACAs, rather than blast increase, may signal the appropriate time for a change of therapy. 

Accelerated phase (AP) should be treated as a high-risk disease. Table 3 shows updates of the current options for managing BC [10,52].

Some form of remission or a return to a second CP (CP2) is recommended prior to transplantation. Transplantation in BC is not advised.

## 7. Treatment Discontinuation and Treatment-Free Remission

Treatment discontinuation in stable DMR, for achieving TFR and possibly cure, is a new goal of CML treatment. DMR is defined as a molecular response of MR^4^ or deeper. In DMR, progression is extremely rare [16]. Benchmark times, when DMR can be expected, have been determined in long-term clinical TKI trials and are summarized in Table 4 [2,16,21,22,27].

Since the initial reports by Rousselot in 2007 [56] and Mahon in 2010 [57], numerous reports on treatment discontinuation have followed. Table 5 lists a selection of 21 studies that include close to 3000 patients. Relapse-free remission at two years ranges around 50% (33–72% at 0.5–10 years).

Durations of DMR and of TKI treatment have been determined as the most important predictors of successful TFR. Low-risk disease and treatment with IFN may also be beneficial [57,80]. EURO-SKI, which with 755 patients is the largest study, reports a TFR rate of 49% at two years [58]. Longer durations of treatment and of DMR correlate with higher TFR rates. The DESTINY Study from the UK indicates that TFR may be more successful if TKI dosage is first reduced before being completely stopped [77]. It is noteworthy that treatment discontinuation in routine care shows similar, if not higher, TFR rates [78], Richter, ESH 2020. This has been associated with longer treatment duration outside clinical trials and more frequent use of 2G-TKIs. However, due to their frequent and serious long-term toxicity, 2G-TKIs are not generally recommended by the ELN for the faster achievement of DMR. Exemptions might be younger patients with low- or intermediate-risk disease, patients for whom TFR is a high priority, and women who wish to become pregnant.

Loss of MMR indicates failure after TFR [61]. After the restart of treatment, about 95% of patients regain pre-discontinuation levels of response. Sources of relapse are thought to be residual BCR-ABL transcripts and the persistence of BCR-ABL1 in the genome [60,81]. In a genomic analysis of 42 patients in DMR, those with a negative PCR in both RNA and DNA had a stable TFR rate of 80–100%, compared to 57–67% in those with a negative PCR in RNA and a positive PCR in DNA, and 20% in those with a positive PCR in both RNA and DNA [81]. 

Most relapses occur early within the first 6 months, but late relapses have been reported in 15% of cases up to 6 years after discontinuation [78].

After relapse, a second treatment discontinuation can be tried, but the success rate is lower [79].

Mandatory, minimal, and optimal requirements for successful treatment discontinuation have been proposed by the European LeukemiaNet [10] (Table 6). 

## 8. Pregnancy

All TKIs are potentially teratogenic and should not be given during pregnancy [82,83]. Since TKIs may be secreted in breast milk, TKI treatment during breast feeding is not recommended [84]. Sperm quality has been found to be unchanged after treatment with TKIs [85].

## 9. Conclusions

Since currently only 15–20% of patients reach stable TFRs, and most patients are at risk of life-long TKI treatment, the amelioration of chronic low-grade side effects is important. The projection for the next 5 years is that more patients will reach TFR through longer treatment duration or treatment with newer more efficacious drugs at a good quality of life, that less patients progress to BC, and that we can address a cure of CML more definitely in a cost-effective manner.

## Figures and Tables

**Figure 1 jcm-09-03671-f001:**
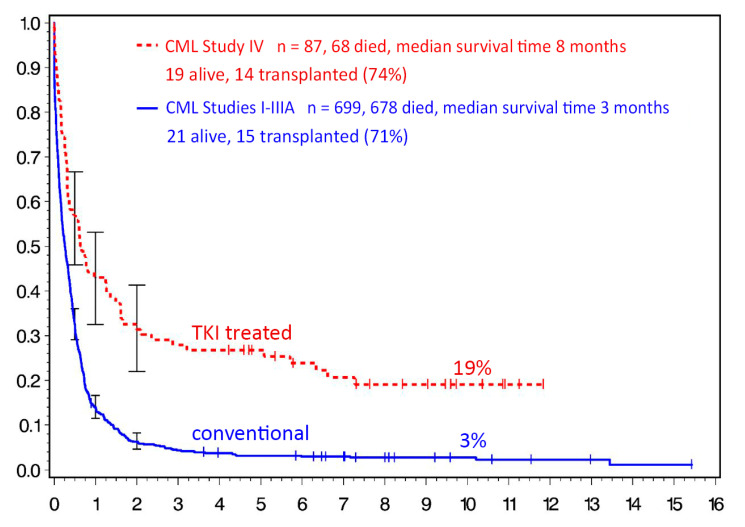
Survival after blast crisis (BC) with conventional (blue) versus tyrosine kinase inhibitor (TKI) therapy (red). German CML Study Group experience, updated 2020 (M.Lauseker). Ten-year survival was 19% (68 of 87 died) after TKI, and 3% after conventional therapy (678 of 699 died); 74% of living TKI-treated BC patients (14 of 19) and 71% of living chemotherapy-treated BC patients (15 of 21) had been transplanted.

**Figure 2 jcm-09-03671-f002:**
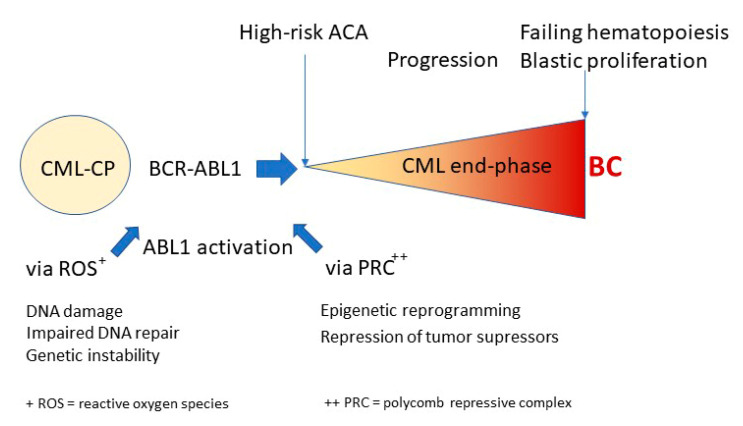
Model of progression of CML from chronic phase (CP) to end-phase and BC. Suggested modulators are reactive oxygen species (ROS) [46] and the polycomb repressive complex (PRC) [48]. Elimination of BCR-ABL1 should stop, or reduce, progression to BC.

**Table 1 jcm-09-03671-t001:** Prognostic score at baseline and comparison of outcome with Sokal and ELTS scores [11,12].

(A) ELTS score calculation.						
Calculation			Definition of Risk Groups			
0.0025 × (age/10)^3^ + 0.0615 × spleen size + 0.1052 × peripheral blood blasts + 0.4104 × (platelet count/1000)^−0.5^			Low risk: < 1.5680 Intermediate risk: 1.5680–2.2185 High risk: > 2.2185			
**(B) Risk strata proportions and outcome.**						
	**Low Risk**		**Intermediate Risk**		**High Risk**	
***n*** **= 5154**	**Sokal**	**ELTS**	**Sokal**	**ELTS**	**Sokal**	**ELTS**
%	38	55	38	28	23	13
10-year OS	89%	88%	81%	79%	75%	68%
6-year LRD	3%	2%	4%	5%	8%	12%

ELTS: EUTOS score for long-term survival, considering leukemia-related death (LRD) and overall survival (OS). Age is given in years; spleen size in cm below the costal margin and measured by palpation (maximum distance); blasts in percent of peripheral blood differential; platelet count as 10^9^/L. All values are pre-treatment. To calculate Sokal and ELTS scores, go to http://www.leukemia-net.org/content/leukemias/cml/elts_score/index_eng.html.

**Table 2 jcm-09-03671-t002:** Milestones for treating chronic myeloid leukemia (CML) expressed as % BCR-ABL1 on the International Scale.

	Optimal	Warning	Failure
Baseline	NA	High-risk ACA, high-risk ELTS score	NA
3 months	≤10%	>10%	>10% if confirmed within 1–3 months
6 months	≤1%	>1–10%	>10%
12 months	≤0.1%	>0.1–1%	>1%
Any time	≤0.1%	>0.1% loss of ≤ 0.1% (MMR) *	>1%, resistance mutations high-risk ACA

* Loss of MMR (BCR-ABL1 > 0.1%) indicates failure after TFR. NA: not applicable; ELTS: EUTOS score for long-term survival; ACAs: additional chromosomal aberrations; MMR: major molecular remission.

**Table 3 jcm-09-03671-t003:** BC management.

Prevention of BC	Effective treatment, elimination of BCR-ABL1
High-risk ACA	Observe closely, intensify treatment
AP	Treat as high-risk CML
Primary BC	Start with imatinib Assess for allo-SCT, initiate donor search
Resistance to 2G-TKI	Ponatinib
Failure to ponatinib	Early allo-SCT recommended
Progress to BC	Attempt at return to CP2 * For myeloid BC: AML regimens + TKI [53,54] For lymphoid BC: ALL regimens + TKI [55] After CP2: allo-SCT without delay

* CP2 = second chronic phase; AP = accelerated phase; SCT = stem cell transplantation; 2G-TKI = 2nd generation TKI; AML = acute myeloid leukemia; ALL = acute lymphoblastic leukemia.

**Table 4 jcm-09-03671-t004:** Benchmark times for DMR (MR^4^ and MR^4.5^) after 5 and 10 years of treatment with imatinib, nilotinib, and dasatinib.

Study		5 Years (%)	10 Years (%)
CML study IV *	Imatinib MR^4^	68	81
	Imatinib MR^4.5^	53	72
ENESTnd **	Nilotinib MR^4^	66	73
	Nilotinib MR^4.5^	54	64
	Imatinib MR^4^	42	56
	Imatinib MR^4.5^	35	45
Dasision ***	Dasatinib MR^4.5^	42	NA
	Imatinib MR^4.5^	33	NA

* imatinib (*n* = 1442), ** nilotinib 300 mg twice daily (*n* = 282), imatinib 400 mg daily (*n* = 283), *** dasatinib 100 mg once daily (*n* = 259), imatinib 400 mg daily (*n* = 260). DMR rates of these trials cannot be directly compared, owing to different methods of trial evaluation. The larger difference between responses to imatinib and 2G-TKI in ENESTnd compared to Dasision correlates with a higher drop-out rate of imatinib-treated patients (50%) compared to nilotinib-treated patients (40%). NA: not available.

**Table 5 jcm-09-03671-t005:** Selected TKI-discontinuation studies.

Study	TKI	Min. Treatment Duration (Years)	*n*	Depth of MR	Min. Duration of MR (Years)	RFS with at Least MMR	References
Euro-SKI	IM	3	755	MR^4^	1	49% at 2 years	Saußele et al., 2018 [58]
STIM	IM	2	100	MR^5^	2	37% at 10 years	Etienne et al., 2017 [59] Update at ESH 2019
TWISTER	IM	3	40	MR^4.5^	2	45% at 42 months	Ross et al., 2013 [60]
A-STIM	IM	3	80	UMRD	2	64% at 23 months	Rousselot et al., 2014 [61]
KID study	IM	3	126	MR^4.5^	2	58% at 2 years	Lee et al., 2016 [62] Update Zang 2018 ASH a. 4252 [63]
STIM2	IM	2	200	MR^4.5^	2	46% at 2 years	Nicolini et al., 2018 [64] ASH a. 462
ISAV	IM	2	112	UMRD	1.5	52% at 22 months	Mori et al., 2015 [65] Update at ASH 2018 a. 461 [66]
STOP 2G-TKI	Dasa/Nilo	2	60	MR^4.5^	2	ca. 55% at 4 years	Rea et al., 2017 [67]
DADI	Dasa 2nd line	ND	63	MR^4^	1	49% at 6 months	Imagawa et al., 2015 [68]
NILST	Nilo	2	87	MR^4.5^	2	59% at 1 year	Kadowaki et al., 2016 ASH a. 790 [69]
TRAD	IM/Dasa	3	75	MR^4.5^	2	58% at 6 months	Kim et al., 2016 ASH a. 1922 [70]
Dasfree	Dasa	2	84	MR^4.5^	1	46% at 2 years	Shah et al., 2019 [71] Update at ESH 2019
ENESTop	Nilo 2nd line	3	126	MR^4.5^	1	58% at 4 years	Hughes et al., 2016 ASH a. 792 [72]
STAT2	IM/Nilo	2	96	MR^4.5^	2	68% at 1 year	Takahashi et al., 2018 [73]
ENESTfreedom	Nilo	2	190	MR^4.5^	1	52% at 4 years	Hochhaus et al., 2017 [74]
D-STOP	IM/Dasa	ND	54	MR^4^	2	63% at 1 year	Kumagai et al., 2016 ASH a. 791 [75]
Spanish study	IM/Nilo/Dasa	3	236	MR^4.5^	2	64% at 4 years	Boluda et al., 2018 ASH a. 47 [76]
DESTINY	IM/Nilo/Dasa	6.9 (median)	125	MR^4^	3	72% at 3 years	Clark et al., 2019 [77]
Routine Care	TKI	7.1	128	MR^4^	4	67% at 2.9 years	Rousselot et al., 2020 [78]
Swedish CML-Registry	TKI (53% IM)	7.7	131	DMR	2.9 (median)	61% outside, 35% inside a study at 2 years	Richter, ESH 2020
RE-STIM	(2nd stop)	3.1 (median)	106	MR^4.5^	1.7 (median)	33% at 4 years	Legros et al., 2017 [79] Update at EHA 2019
**Total: 21**			**2974**			**33–72% at 0.5–10 years**	

Updated from Hehlmann 2020 [9].

**Table 6 jcm-09-03671-t006:** Requirements for tyrosine kinase inhibitor discontinuation.

Mandatory	CML in first CP only (data are lacking outside this setting)
	Motivated patient with structured communication
	Access to high-quality quantitative PCR using the International Scale (IS), with rapid turn-around of PCR test results
	Patient’s agreement to more frequent monitoring after stopping treatment. This means monthly for the first 6 months, every 2 months for months 6–12, and every 3 months thereafter
Minimal (stop allowed)	First-line therapy or second-line, if intolerance was the only reason for changing TKI
	Typical e13a2 or e14a2 BCR-ABL1 transcripts
	Duration of TKI therapy > 5 years (>4 years for 2G-TKI)
	Duration of DMR (MR^4^ or better) > 2 years
	No prior treatment failure
Optimal (stop recommended for consideration)	Duration of TKI therapy > 5 years
	Duration of DMR > 3 years if MR^4^

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
