# Peer review of "The New ELN Recommendations for Treating CML"

_jcm, 2020, doi:10.3390/jcm9113671_

Round 1
Reviewer 1 Report
This is a useful and clear review of the recently published updated version of ELN recommendations for CML diagnosis and treatment. The Author is an outstanding international leader in the field, as well as the senior author of the reviewed paper. In this manuscript, he provides an insight about the main topics and recommendations, along with some updates.
I have only minor comments and suggestions:
a) it would be useful to recall in the capture of table 1 the reference for data reported in section B
b) the sentence at line 142 ("early transplantation in CP is associated with better outcome") should be detailed more (i.e. better than late transplant? better than TKI? in the latter case a comment about the circumstances of the randomized trial leading to this statement would be required, to avoid misinterpretation about the current role of transplant)
c) table 3: it is unclear the meaning of the label "prevention" as the header of the left column
Author Response
Reviewer 1
Thanks for the comments.
a) References 11 and 12 have been recalled in the capture of Table 1.
b) Early transplantation is associated with better outcome than late transplant.
c) Prevention is not the header of the left column. Lay-out and corresponding wording in the right column will be improved, compare my query regarding lay-out.
Reviewer 2 Report
This is an excellent review on the ELN recommendation for the treatment of CML. The author provides a comprehensive overview of the relevant literature and discusses the most topical points on prognostic scores, treatment milestones, first- and second-line treatment, as well as treatment discontinuation/treatment-free remission.
I have only a few minor comments to address:
line 12: “Generic imatinib is current ...”: article missing, either “imatinib is the current” or “imatinib is a current”
line 99/100 and line 104/105: “recognizes imatinib resistant ...” I would suggest to change the verb to either “binds imatinib resistant ...” or “inhibits imatinib resistant” or “CML patients with mutations x/y/z/ respond to dasatinib/nilotinib.
Author Response
Reviewer 2
Line 12 Generic imatinib is the current....
Lines 99/100 and 104/105: "recognizes" changed to "inhibits". Thanks for the suggestion.
This manuscript is a resubmission of an earlier submission. The following is a list of the peer review reports and author responses from that submission.